# Efficient and Secure WiFi Signal Booster via Unmanned Aerial Vehicles WiFi Repeater Based on Intelligence Based Localization Swarm and Blockchain

**DOI:** 10.3390/mi13111924

**Published:** 2022-11-08

**Authors:** Gehad Abdullah Amran, Shuang Wang, Mohammed A. A. Al-qaness, Syed Agha Hassnain Mohsan, Rizwan Abbas, Eissa Ghaleb, Samah Alshathri, Mohamed Abd Elaziz

**Affiliations:** 1Department of Management Science and Engineering, Dalian University of Technology, Dalian 116024, China; 2College of Software Engineering, Northeastern University, Shenyang 110169, China; 3College of Physics and Electronic Information Engineering, Zhejiang Normal University, Jinhua 321004, China; 4Optical Communications Laboratory, Ocean College, Zhejiang University, Zheda Road 1, Zhoushan 316021, China; 5Department of Computer Science and Technology, Zhejiang University, Hangzhou 310027, China; 6College of Computer and Information Science, Southwest University, Chongqing 400715, China; 7Department of Information Technology, College of Computer and Information Sciences, Princess Nourah bint Abdulrahman University, P.O. Box 84428, Riyadh 11671, Saudi Arabia; 8Department of Mathematics, Faculty of Science, Zagazig University, Zagazig 44519, Egypt; 9Faculty of Computer Science &Engineering, Galala University, Suze 435611, Egypt; 10Artificial Intelligence Research Center (AIRC), Ajman University, Ajman 346, United Arab Emirates; 11Department of Electrical and Computer Engineering, Lebanese American University, Byblos 4307, Lebanon

**Keywords:** chaotic internet of things transports, swarm intelligence localization, blockchain distributed databases, unmanned autonomous vehicles booster, emergency communication vehicles

## Abstract

Recently, the unmanned aerial vehicles (UAV) under the umbrella of the Internet of Things (IoT) in smart cities and emerging communities have become the focus of the academic and industrial science community. On this basis, UAVs have been used in many military and commercial systems as emergency transport and air support during natural disasters and epidemics. In such previous scenarios, boosting wireless signals in remote or isolated areas would need a mobile signal booster placed on UAVs, and, at the same time, the data would be secured by a secure decentralized database. This paper contributes to investigating the possibility of using a wireless repeater placed on a UAV as a mobile booster for weak wireless signals in isolated or rural areas in emergency situations and that the transmitted information is protected from external interference and manipulation. The working mechanism is as follows: one of the UAVs detect a human presence in a predetermined area with the thermal camera and then directs the UAVs to the location to enhance the weak signal and protect the transmitted data. The methodology of localization and clusterization of the UAVs is represented by a swarm intelligence localization (SIL) optimization algorithm. At the same time, the information sent by UAV is protected by blockchain technology as a decentralization database. According to realistic studies and analyses of UAVs localization and clusterization, the proposed idea can improve the amplitude of the wireless signals in far regions. In comparison, this database technique is difficult to attack. The research ultimately supports emergency transport networks, blockchain, and IoT services.

## 1. Introduction

The interest in UAVs has grown in many applications due to developments in manufacturing technology. In addition, UAVs have established various applications and solutions in multiple areas, such as search, rescue, parcel distribution, and telecommunications. As a result, UAVs have appeared as a potential emerging technology in the next decade [1,2]. It is also worth noting that using UAVs in a wide area requires securing database servers and embedding additional devices. In rescue scenarios, for efficient data processing, a high-performance UAV movable network with repeaters and surveillance cameras is required to achieve the rescue mission successfully [3]. New UAV research will likely continue, as all of these experiments will benefit from linking UAVs to the 5G network for improved monitoring and connectivity. There are some studies that explore the obstacles and prospects for developing UAVs using the 5G ground base station (5G-BSs) or emergency communication vehicles. Integrating UAVs and 5G-BSs will increase equipment interference. Some researchers open the door to new techniques to ensure that aerial and land equipment can merge into the same system [4]. There is a shortage of adequate assistance for data transmission in crises and epidemics. For example, the coronavirus outbreak was an issue around the world. UAVs have emerged as a suggested approach to improve the usage of networks in epidemic areas by acting as an alternative transport system integrated with radio signal boosters [5]. One could say that an emergency communication network is essential in the emergency rescue and the event of exceptional circumstances, particularly when the integrated communication system is lost. However, existing approaches lack durability has and have limited space and environment. To address these weaknesses, unmanned aerial vehicles (UAVs), 5G-BSs, UAV signal boosters, and database servers provide an effective wireless coverage area with stability and mobility advantages [6].

This study contributes to utilizing the UAVs embedded with smart thermal cameras to monitor crowds in a specific area that is supported by a wireless signal booster device to strengthen the weak wireless signal in remote areas in emergencies. Figure 1 illustrates the contributions to this paper, which are summarized as follows:

We used UAVs equipped with a thermal camera and signal repeater to identify persons in a specific area and boost the wireless signal in this specific area. The procedure of requesting help through UAVs starts with recognizing indicators in specific regions that will be a gathering point for the people who need help in the case of a request for assistance. UAV cameras employ vision processing technology to recognize the help requesting indicators.The UAVs localization and clusterization scenario is optimized using a swarm intelligence localization (SIL) algorithm.To secure data transmission, blockchain decentralized data is used.

The remaining structure of the paper is as follows. The related work is presented in Section 2. The preliminaries about the swarm method and blockchain cooperative to optimize the wireless signal are introduced in Section 3. Implementation of the system is illustrated in Section 4, and the results are summarized in Section 5. The analysis of the results is shown in Section 6. Finally, the closing and forward-looking comments are in Section 7.

## 2. Related Work

The contribution of previous studies concerning the localization and clustering of UAV applications has recently resulted in detailed research. Fleet operators of UAVs, such as Facebook, Google, and Amazon, aim to transport products and data concurrently by flying drones that follow specified routes and locations. In addition, cellular UAV networks can follow human addresses and travel that can be accomplished by public transit systems [7,8,9,10]. An interesting UAV prototype for low-altitude applications is called the Nokia F-Cell project [11]. In addition, a Huawei Wireless UAV lab under the Digital-Sky project [12,13] involves drones attached to air quality control agencies in Shanghai, China. The gaps in these studies reflected a high cost, no creative form of optimization to identify the target node, and no substantial self-configuration to coordinate drones using a centralized database that may be under attack. Several papers have been published in recent years suggesting novel algorithms for optimizing positioning. Earth observation by UAV uses the convolutional neural network (CNN) for disaster relief in the smart city [14]; the ultra-violet spectrum (UV) [15,16,17] uses the Hough transform process and UVDAR system, which is more efficient and less computationally intensive. There have also been novel experiments using distance-based docking [18,19], blurry approach [20], radar-based high-precision 3D location [21], and UAV relative distances [22,23] of UAVs used to land in environments that have been banned. Other authors proposed a multi-UAV based on a crowd surveillance system that employs UAVs to monitor a group of walking people [24]. Some researchers focused on the security issue of the Internet of Drones Things (IoDT), whereby lightweight blockchain is considered a security solution [25,26]. The combination of cuckoo and Hampel filters [27] is used as a type of blockchain security. Some other studies noted UAV applications in the 5G or 5GB era, from the aspect of wireless communication and its underlying physical characteristics, such as air-to-ground common, energy-efficient channels [28,29]. Some other recent research about UAVs about tasks such as search and rescue, trajectory planning, and artificial intelligence are attractive [30,31]. In addition, it is good to note and compare some other existing work on UAV combined with blockchain [32,33]. Some studies have explored using digital twins in unmanned aerial vehicles to rapidly and efficiently deliver medical resources during an epidemic. The feasibility of DTs in COVID-19 prevention and control is investigated. Algorithms for deep learning (DL) are introduced. The DTs information prediction model is based on an upgraded AlexNet network, the performance of which is evaluated via simulation [34]. Some other studies used UAVs in many applications, for instance, the scenario of a 3D terrain environment represented by triangular mesh data based on the Gaussian factor [35], joint UAV position optimization based on DNS [36], cloud-edge computing [37], crowd systems [38], and safety-critical systems [39]. Other studies are directed towards improving the ability of internal UAVs by optimizing hovering efficiency [40], energy-aware relay [41], oblique photogrammetry with multiple cameras onboard UAVs [42], or environmental sense optimization for UAVs [43,44]. Table 1 highlights existing work along with the limitations. All previous work noted above relies primarily on the representation of nature. The most current study assumes a fixed or non-modulated environment. The theory of early weather prediction is not adopted. Moreover, much of the reviewed work does not care about the decentralization of data and does not care about fixed IoT data backups that are vulnerable to infiltration.

## 3. Preliminaries

This section contains theoretical information about using UAVs to boost wireless signals in isolated cities. SIL, as a specific form of the swarm, considers an ideal solution to the localization problem. Blockchain is regarded as data sharing with decentralization advantages and tamper-resistant functionality. SIL and blockchain must work together to ensure a synchronized, secure and efficient wireless signal. As defined in the following subsections, the SIL strategy focused on coordination in decision-making between group and individual intelligence. In comparison, the blockchain is distinct from the centralized system, as all transactions are maintained in blocks of entities within the blockchain, which has made blockchain popular in digital currency implementations and has opened the door to comprehensive research on the Internet of Things. The assumptions of the proposed scheme are:The UAVs could be used to boost wireless signals in far cities during emergency cases.The swarm is an ideal solution to the localization problem.Blockchain technology is an appropriate data sharing technology for decentralization advantages.SIL and blockchain could work together to ensure a synchronized signal booster.

### 3.1. Swarm Intelligence-Based Localization (SIL)

Swarm intelligence localization is an intelligent optimization algorithm based on the swarm intelligence model inspired by animal behavior, such as in bird colonies. The suggested position is determined based on the particle’s direction and the location of the neighbor [45]. The follower-UAVs are searching for the address of the leader-UAV in a particular range area. Follower-UAVs are working together to solve this localization problem. Each follower-UAV measures the local position based on the best private position (Pbest) and the best general swarm position (Gbest) in the search space and informs each other of the measured location, and they can choose the best general location for the leader-UAV. All UAVs have the memory to store their best position and velocity value after each iteration round. The challenge with optimization is to find a better position to find the right location; this level of intelligence is utterly unreachable for any member of the swarm, but it can be achieved by coordination among the swarm members [46].

### 3.2. Computational Complexity

As seen in Figure 2, the swarm comprises three follower-UAV candidates represented by *X*_i_(t), where “*X*” is the location of the UAV, “i” is the number of iterations, and “t” is the function of time for each search space. In addition to the position, there is a velocity for each part, denoted by V_i_(t). The trajectory of *X*_i_(t) and *V*_i_(t) are the same, where the velocity term represents the follower-UAV motion in the direction and time. Each swarm candidate has a recollection of its own best location *P*_i_(t), and this is the best experience of a partial “i”. Often, the swarm has the best-shared among the participants of the swarm G(t) without the “i” index, since it belongs to the entire swarm. The follower-UAV travels into a new position and velocity using all three of the elements represented by *P*_i_(t), G(t), and their updated position and velocity denoted by *V*_i_(t + 1) and *X*_i_(t + 1). Equations (1)–(5) fully explain the mathematical complicity behind SIL optimization.
(1)$$V_{ij\;}\left(t+1\right)=Inertia\;term+Cognitive\;component+Social\;component\;$$
(2)$$Inertia\;term=W\;\cdot \sum_{i=0}^{n}\sum_{j=0}^{m}V_{ij\;}\left(t\right)$$
(3)$$Cognitive\;component=R1\;\cdot \;C1\;\cdot \sum_{i=0}^{n}\sum_{j=0}^{m}(P_{ij\;}\left(t\right)-X_{ij\;}\left(t\right))\;\;\;\;\;\;\;\;\;\;\;\;\;\;\;\;\;\;\;\;\;\;\;\;\;\;\;$$
(4)$$Social\;component=R2\;\cdot \;C2\;\cdot \sum_{i=0}^{n}\sum_{j=0}^{m}(G_{j\;}\left(t\right)-X_{ij\;}\left(t\right))\;\;\;\;\;\;\;\;\;\;\;\;\;\;\;\;\;\;\;\;\;\;\;\;\;\;\;\;\;\;\;$$
(5)$$X_{ij}\;\left(t+1\right)=\sum_{i-0}^{n}\sum_{j=0}^{m}X_{ij}\;\left(t\right)+V_{ij}\;\left(t+1\right)\;$$
where:

*V*ij(t), velocity per time;

*X*ij(t), position per time;

*P*ij(t), best private position for follower-UAV;

*G*j(t), the best general position towards leader-UAV;

*V*ij(t + 1), new velocity per time;

*X*ij(t + 1), new position per time;

*W*, real value coefficient;

*C*1,2, acceleration coefficient;

*R*1,2, random number uniformly distributed in the range of 0 to 1.

**Figure 2 micromachines-13-01924-f002:**
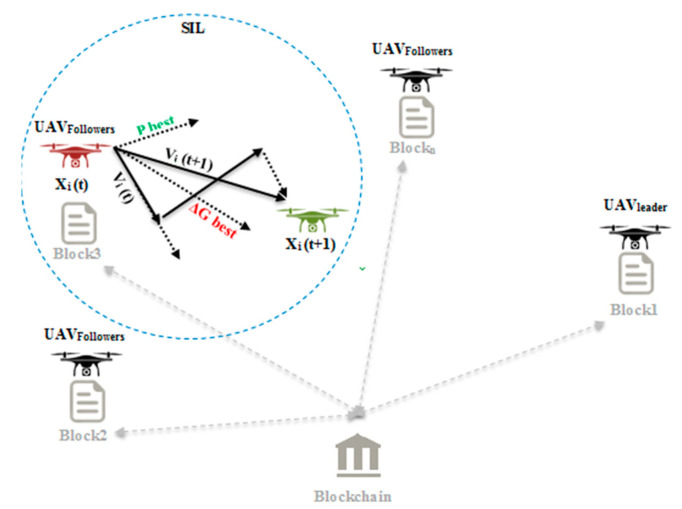
Swarm intelligence base localization (SIL) model. Drone movement from *X*i(t) to *X*i(t + 1) dependent on *V*i(t), Pbest, and Gbest.

### 3.3. Blockchain: Decentralization of Big Data

Blockchain is the transfer of functions among all the units concerned rather than the central unit. A copy of the transaction ledger [47] is given to each blockchain drone. In the decentralized environment, the network functions under a peer-to-peer (user-to-user) policy. The blockchain protocol has four characteristics:All nodes are supposed to have the same output.All non-fault nodes have a value.There is resistance during recovery of the failure node.Blocks in the blockchain network regularly search for self-audit every 10 s [48].

The structure of the blockchain is clearly shown in Figure 3. Blockchain is a continuously expanding collection of records called blocks. The blocks are connected and guarded by cryptography. Each block contains the hash tree (root, node, and leaf), timestamp, and transaction details. If UAVs discover wireless signal weakness, the transaction details will change the information within the relevant block. All blocks comply with the latest upgrade created by the leader-UAV block.

## 4. Proposed System Implementation

The simulated system consists of four UAVs configured as one leader-UAV and three follower-UAVs, as shown in Figure 4. Simulation software was written to handle the network using MATLAB simulator. The proposed software consists of two technologies, the first is SIL optimization, and the second is blockchain for each UAV. The information in UAV blocks changes every 10 s. Leader-UAV binds to follower-UAVs and sends hash information within the data block. All UAV blocks are following over the blockchain communication protocol. All UAVs float over the far region inside the emergency wireless network to conduct a signal-boosting mission. The UAV scheme was designed by merging the network layer and the blockchain layer, as illustrated in Figure 3 and below:Network layer: In the network layer, UAVs are divided into leader-UAV and follower-UAVs. In the case where a drone investigated and found a weak wireless signal, the drone becomes a leader-UAV and broadcasts its position in swarm space. Then, the leader-UAV interacts with other UAVs, who become follower-UAVs. During the contact, the follower-UAVs affirm reception of the location of the leader-UAV. At the same time, follower-UAVs coordinate with each other using the SIL swarm technique to travel into the weak signal zone. UAVs boost the signal region after the operation. A decentralization database is a secure connection database between UAV swarms.Blockchain layer: The blockchain layer is responsible for protecting all transactions preserved by block actors within the UAV structure in the blockchain layer. In particular, the UAV block is created within the platform program. Both blocks are free to gather data in the blockchain from the leader-UAV block after modifying the location information and upgrading their inside block hash tables. Each UAV block in the blockchain network creates a Merkle tree that depends on the number of transactions. The transections here show the data on velocity and position. Within the umbrella of the blockchain, the evolved version of the Ethereum protocol from the point of view of decentralization selectivity, *G*_j_(t) is picked as the best-chosen position for UAVs swarm, and *P*_ij_(t) is the best position for a particular UAV; *G*_j_(t) and *P*_ij_(t) together determine which UAV token to transfer.

**Figure 4 micromachines-13-01924-f004:**
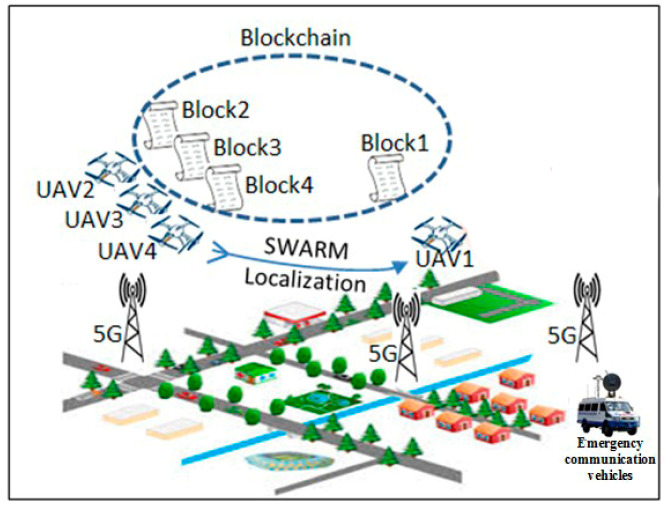
Demonstration of UAVs based on SIL and blockchain during emergency.

### UAV Network Architecture

The proposed SIL is inspired by the social behavior of bird flocking. The method is simulated in MATLAB 0.9 U.S, where all particles travel through a computational search space to find an optimal solution. The virtualized area of the simulation network is 1000 × 1000 m, where the UAVs are evenly distributed. The UAV transmission range is roughly 250–300 m. 64 UAV nodes (a suitable number of nodes to show good resolution after several trials) are uniformly distributed in this region. Among these nodes, 1 and 63 UAV nodes are the leader and follower nodes, respectively. These nodes are randomly spread. The correction factor is 2, the inertia factor between (0–1), the blockchain number 64, and the highest iteration number 50. The simulation was performed on a Windows 10 OS [49] with four intel core i5-2430M central processing units (CPU) of ~2.4 GHz and 8GB of DDR4 random access memory. Simulation parameters are summarized in Table 2. Algorithm 1 indicates the structure of the scheme suggested. The algorithm starts with initializing all the particle swarms (steps 1–4). The leader-UAV forwards the goal location and the velocity of the environment (Steps 5–7). Based on the leader-UAV role, the best UAV position, and the best global address, the particle solution is changed (Steps 8–16) with each iteration. Particle velocity and location are measured and modified using Equations (1)–(5). As in Algorithm 1, blockchain hash information is updated as an entity in a decentralized system throughout the iteration.
**Algorithm 1**. Localization and clustering**Input:** S, *V*ij(t), *X*ij(t), *W*, *C*1, 2, *R*1, 2, *N*, *K*.**Output:***P*ij(t), *G*j(t), *V*ij(t + 1), *X*ij(t + 1)./* Initialization*/
//Generate the initial S, Vij(t), and Xij(t) of dimensions where each particle’s position and velocity are randomly generated in the searching space.
1: Initialize *S*, *W*, *V*ij(t)
2: for ∀ UAVs ∈ *S* (1≤ i ≤ N) do
3: Initial *X*ij(t)
4: end for/* Leader-UAV*/
//Generate the target position and velocity in the searching space.
5: If the thermal signature (is true) then
6: UAV position send to update in blockchain
7: end if/* Main loop*/
8: while (Stopping criteria (false)) do
9: for each k do/* Computation*/
//Calculate the suitable position for follower-UAV *X*ij(t + 1) by using current *X*ij(t) and *V*ij(t + 1) = {*P*ij(t), *G*j(t)} based on Equations (1) and (5).
10:Calculate *V*ij(t + 1) using Equation (1)
11:if *V*ij(t + 1) > Vmax then
12:Set *V*ij(t + 1) = Vmax 
13:end if 
14:if *V*ij(t + 1) ≤ Vmax then
15:Set *V*ij(t + 1) = Vmin
16:end if /* Blockchain*/
//Update the velocity and position of the current generation.
17:for ∀ UAVs ∈ S (1≤ i ≤ N) do
18:Update ∀ 10 sec the position *X*ij(t + 1) and *P*ij(t)
19:Conform receive all hash UAVs updated position
20:end for/* Output*/
//Private best position Pij(t) for UAV, Global best position Gj(t) for swarm, next velocity *V*ij(t + 1), next position *X*ij(t + 1).
21:Calculate *P*ij(t), *G*j(t), *V*ij(t + 1), *X*ij(t + 1).
22:the end for
23:end while

## 5. Results

A numerical simulation was carried out in MATLAB to confirm the validity of the analytical model. Figure 5 demonstrates transition probabilities for UAVs that rely on swarm and blockchain principles. The processing time of the UAVs to adjust their position corresponding to the leader-UAV position is about 5–6 s, depending on the location of the follower-UAVs, at the time of randomly selecting the first implementation. As shown in Figure 6, the scenario starts with assembling 64 UAVs as a matrix. The UAVs’ locations in the network are in random positions, where all UAVs must change direction depending on the leader-UAV. The follower-UAVs are distributed randomly at an unknown address within the network at the start of the deployment. The leader-UAV shares its position with the rest of the follower-UAVs. Follower-UAVs confirm receipt of the leader-UAV location, and this confirmation was achieved by blockchain procedure. Then, the follower-UAVs go to the leader-UAV spot, following the swarm strategy, to get the fastest way there. Figure 5 and Figure 6 display the phases of UAV motions from random positions to one position, and a 3D diagram shows the direction of all the red nodes towards the green node, represented by the leader-UAV. The number of leader-UAVs is not analyzed here, as UAVs are not affected by the number of leader-UAVs. The blockchain usage in the proposed structure is handled as follows. The leader-UAV sends a block consisting of the current hash information and the previous hash address, timestamp, and other information. This information is frequently compared in all decentralization blocks, and in the event of a signal weakness alert in any UAV block, UAVs transfer directly to the location by following the location information in that hash address.

## 6. Performance Analysis

### 6.1. Localization Analysis and Discussion

Figure 7 depicts the implementation of four representative nodes in the proposed UAV network. The red, green, and blue lines indicate the follower-UAV movements, and the green drone marks the leader-UAV target node. The figure clearly shows that all the lines point to the anchor node. The location error is inversely proportional to increasing the number of nodes, where the best general swarm position *G*_j_(t) and the best personal position *P*_ij_(t) values both contribute to correcting the position of the next follower-UAV *X*_ij_(t + 1). In the scenario of 64 follower-UAVs reaching the leader-UAV in 5–6 s during 50 iterations, the increasing number of nodes would minimize the iterations and time consumption to get the target node. An adequate number of anchor nodes may help to improve the coordination system. There are also some unpredictable values in (18–20) iterations for several nodes that cause them to miss their target and re-correct their location to return to the time-consuming global swarm path. In addition, having more leader-UAVs would boost the UAVs’ network behavior. The number of hops and the destination among UAVs to reach the swarm mainline are inversely proportional. The nearest node is greater than the global swarm value, and the simulation performance of the proposed method shows less localization error. All follower-UAVs reach the leader-UAV position after (~26) iterations. Although the swarm intelligence location (SIL) algorithm follows partial swarm optimization (PSO), which works on a known location, SIL depends on an unstructured node network that depends on the unknown location of neighbor nodes. SIL’s flexibility makes it more ideal for the Internet of Drone Things (IoDT), which relies on an unpredictable location approach, unlike other state-of-art bio-inspired algorithms, such as dragonfly and penguin.

### 6.2. Security Analysis and Discussion

According to the findings, using blockchain as a decentralized database in the Internet of Things applications will be beneficial in reducing time consumption. The Internet of Things relies on working with different devices, which need separate memory for each one that obeys the blockchain protocol. In reality, it is not easy for an attacker to attack all devices in the system individually, and there is no point in including malicious software in one of the device blocks, as the device is continuously upgrading data. As explained above, in the proposed model, blockchain synchronized interconnected blocks for the UAV network depend on the idea of updating and confirmation. Hence, the blocks communicate directly with each other without any third party. In general, each block contains two parts, the block header (hash list) for authentication and the block body (transaction) for data transmission [50], represented by the UAV location and the isolated region of the suggested proposal. As seen in Figure 8, the upgrade duration of the blockchain protocol is predetermined. In case the UAV detects a poor-signal zone, the UAV will change its data on its current position, and its condition will convert from a UAV to a leader-UAV. After 6.47 s, the remainder of the UAVs will confirm that they received the latest location. Then, follower-UAVs move to their expected location and aid by boosting the wireless signal in the target area. The blockchain performance analysis and storage are shown in Figure 9. The leader-UAV sends the location information as an encrypted block utilizing related and previous hashes to the decentralized database, wherein blockchain is used to communicate the location with follower-UAVs.

### 6.3. Blockchain Threat Modeling

The proposed scheme for the security threats is described clearly in Figure 10. The suggested security procedure includes four levels that encompass blocks initializing what security means for the whole system, eliciting threats, analyzing the treating risks, and evaluating how countermeasures lead to the achievements of security objectives.

## 7. Conclusions and Outlook

This research discusses the localization and assembly of UAVs during emergency scenarios utilizing SIL and blockchain theories to improve the rescue signal in remote places. If a UAV detects a low-signal region inhabited by a group of people, it communicates its position data to surrounding UAVs to collect it to aid with signal strengthening. Whereas the UAVs use the SIL theory to gather data, the geographical information of the UAVs is stored in a decentralized database that is protected from hacking and manipulation using the blockchain method. It was discovered that increasing the number of UAVs reduces execution time, redundancy, and localization inaccuracy. It was also noticed that syncing data via blockchain may save time. In the future, a different localization algorithm inspired by nature may offer using the blockchain technique to produce alternative outcomes that could be discussed in the laboratory.

## Figures and Tables

**Figure 1 micromachines-13-01924-f001:**
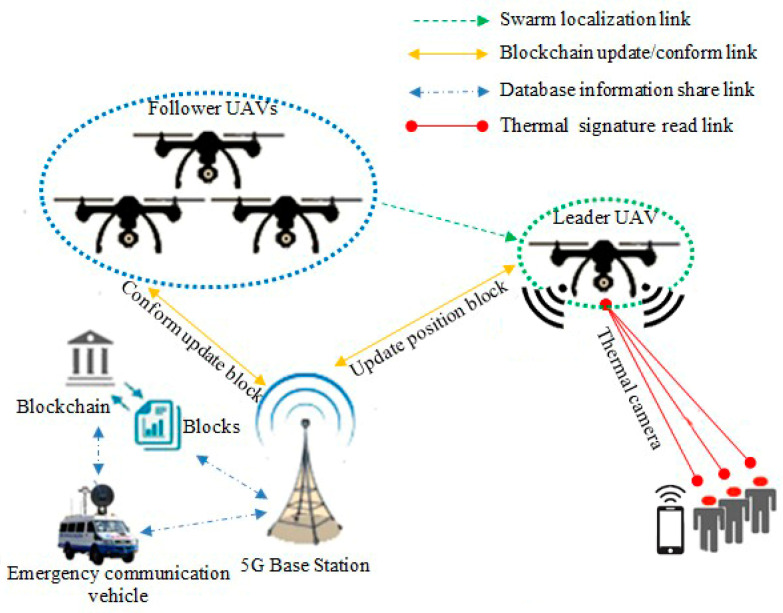
Side layout of a suggested boosting transmission model for UAV movement in a weak coverage area. The leader–follower strategy used swarm and blockchain approaches.

**Figure 3 micromachines-13-01924-f003:**
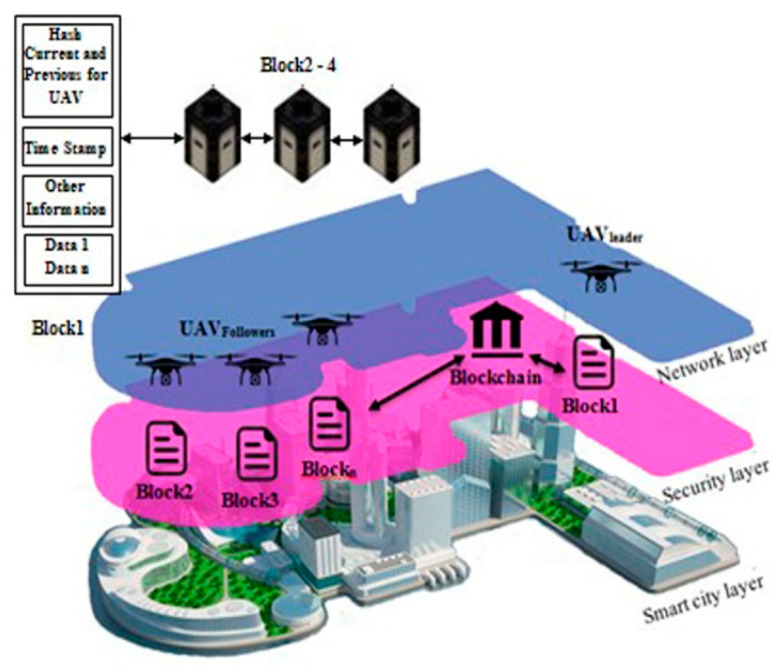
Layer representation of a blockchain. Drone location is updated in leader–UAV block and sends the update to the follower–UAV blocks using blockchain routing. The follower–UAVs conform to leader–UAV received updates.

**Figure 5 micromachines-13-01924-f005:**
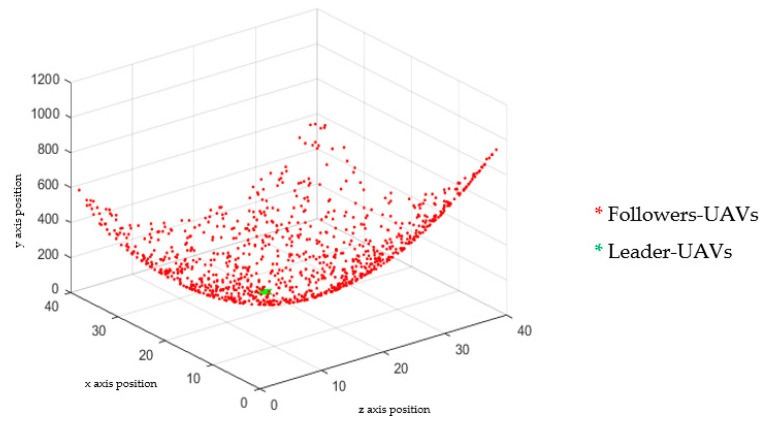
Three-dimensional MATLAB simulation diagram showing the result of directed follower-UAVs toward the leader-UAV location. The final result was estimated after 50 iterations.

**Figure 6 micromachines-13-01924-f006:**
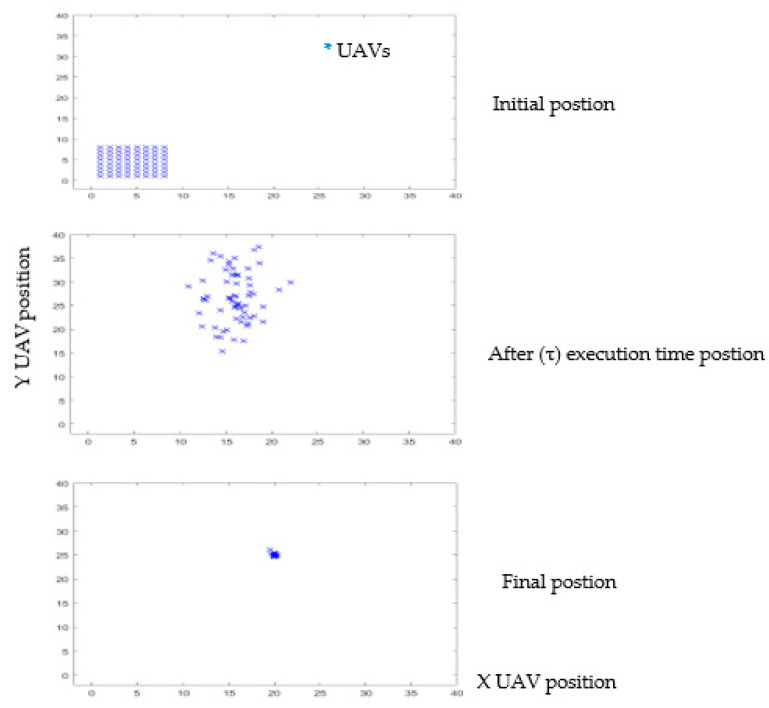
MATLAB execution stages for localization optimization of 64 nodes of UAVs and 50 iterations. Here, Leader–UAV works as a reference point to estimate 64 follower-UAV node locations. The proposed SIL algorithm achieved less estimation location error per time.

**Figure 7 micromachines-13-01924-f007:**
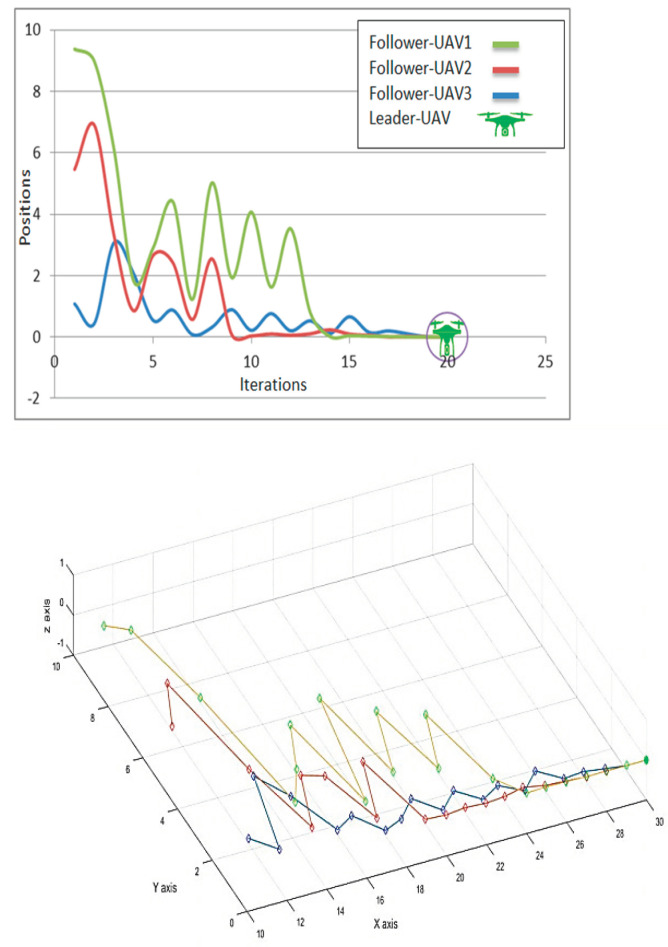
Leader–UAV and follower–UAV trajectories comparison depend on position and velocity. Colored lines (green, red, blue) correspond to specific follower-UAVs and the green drone represents the leader–UAV location.

**Figure 8 micromachines-13-01924-f008:**
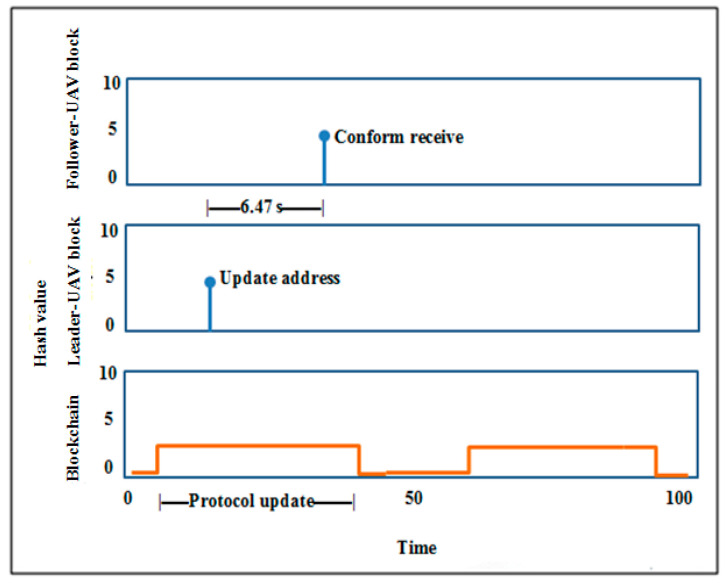
Blockchain timing diagram. Leader-UAV/follower-UAV block relationship.

**Figure 9 micromachines-13-01924-f009:**
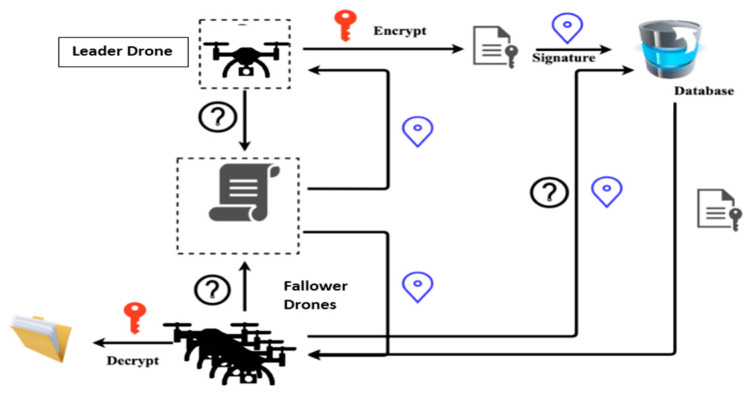
Blockchain performance analysis and data storage.

**Figure 10 micromachines-13-01924-f010:**
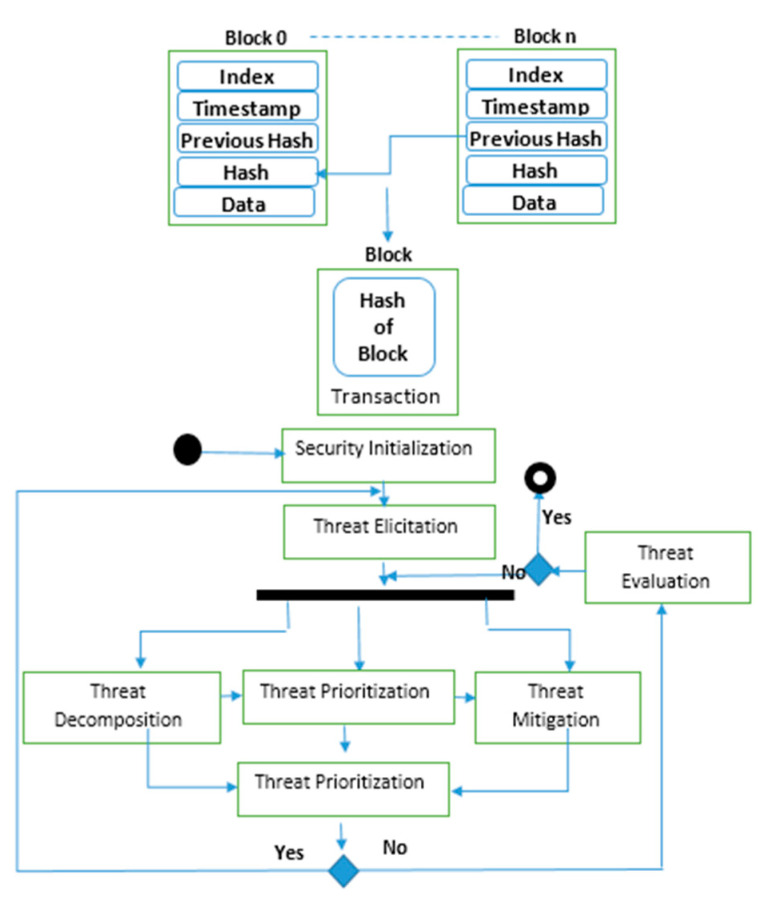
Blockchain threat modeling and process analysis.

**Table 1 micromachines-13-01924-t001:** The prior studies along with their limitations.

References	Limitation
[8,9,10,11,12]	High cost. No creative form of optimization to identify the target node. No substantial self-configuration to coordinate drones using a centralized database that may be under attack.
[13,14,15,16,17,18,19,20,21,22,23,24,25]	Efficiency. QoS.

**Table 2 micromachines-13-01924-t002:** Simulation parameters.

Parameter	Value
Network simulator	MATLAB
Network area	1000 × 1000 m
UAV transmission range	250–300 m
Swarm size	64
Correction factor	2
Inertia	1
Number of leader-UAVs	1
Number of follower-UAVs	63
Number of blocks	64
Localization algorithm	SIL
Clustering protocol	Blockchain
Maximum number of iterations	50

S = swarm size, *X*ij(t) = inertia component, *V*ij(t) = random initial velocity, W = inertia coefficient C1, 2 = acceleration coefficient, R1, 2 = uniform distributed number U (0, 1), N = population size, K = number of iteration, *P*ij(t) = cognitive component, *G*j(t) = social component, *V*ij(t + 1) = next position, next step velocity, *X*ij(t + 1) = next position.

## Data Availability

The data used to support the findings of this study are available from the corresponding author upon request.

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
