# Peer review of "Efficient and Secure WiFi Signal Booster via Unmanned Aerial Vehicles WiFi Repeater Based on Intelligence Based Localization Swarm and Blockchain"

_micromachines, 2022, doi:10.3390/mi13111924_

Round 1

Reviewer 1 Report

This submission contributes to investigating the possibility of using a wireless repeater placed on a UAV as a mobile booster for weak wireless signals in isolated or rural areas in emergency situations, and that the transmitted information is protected from external interference and manip-34 ulation. The working mechanism is as follows, one of the UAVs detects a human presence in a pre-determined area by the thermal camera and then directs the UAVs to the location to enhance the weak signal and protect the transmitted data. The methodology of Localization and clusterization of the UAVs is represented by Swarm Intelligence Localization (SIL) optimization algorithm. Mean-while, the information sent by UAV is protected by Blockchain technology as a decentralization database

The discussions look valuable. However,

1.This submission has not sufficiently clarified the novelty of the proposed approach. No equations are provided as a scientific submission.

2. This submission misses discussing a few relevant works, such as

- "Energy-Aware Relay Optimization and Power Allocation in Multiple Unmanned Aerial Vehicles Aided Satellite-Aerial-Terrestrial Network", IEEE Systems Journal, Early Access, pp. 1-12, March 2022, DOI: 10.1109/JSYST.2022.3147491

- “Indoor Fingerprinting Localization and Tracking System Using Particle Swarm Optimization and Kalman Filter”, IEICE Transactions on Communications, vol. E98-B, no. 3, pp.502-514, Mar. 2015

- “Efficient Indoor Fingerprinting Localization Technique Using Regional Propagation Model,” IEICE Transactions on Communications, vol. E97-B, no. 8   pp.1728-1741, August 2014

- "Joint UAV Position Optimization and Resource Scheduling in Space-Air-Ground Integrated Networks with Mixed Cloud-Edge Computing", IEEE Systems Journal, vol. 15, no. 3, pp. 3992-4002, September 2021, published December 2020, DOI: 10.1109/JSYST.2020.3041706, 

Author Response

Dear reviewer,

       Please find our response letter in the attached document.

Thank you 

Reviewer 2 Report

The paper attempts to apply SIL and blockchain in a Swarm drones communication for search and rescue operations. The drones in this use case are considered to be a booster. The authors have interesting ideas and results, however, the paper needs to be thoroughly revised to increase its quality and especially to deliver the contributions.

1. First of all I think the title is misleading. The authors' work just considers the 5G signal booster to imagine the use of swarm drones, but the authors didn't include any of the communication aspects (5G parameters and requirements, booster/repeater technical capacity and demand, etc) in the localization and data exchange. As I see it, the proposed schemes can work in any network and cellular technology. So I propose revising the title, amongst others, to include the word "swarm drones" as it is one of the cosu of the study. 

2. My biggest concern in the study is the introduction and related works. In the introduction and related works, the authors just introduce the diverse use of UAVs but not the concrete research gaps.  In current writing, the ideas of using swarm with SIL and Blockchain technology seem like pop-up ideas and are not justified academically. 

3. Authors mention several obstacles to developing UAVs using 5GBS, such as increased interference, but this is not discussed in the obtained results or methodology. The lack of durability is not clear to the readers. Author's also mentioned about "early weather prediction is not adopted", which I don't see brings any value to the work unless discussed.

4. "Using UAVs equipped with a thermal camera, and signal repeater to identify persons in a specific area and boost the wireless signal in this specific area..." is definitely not the authors' contribution, as many have proposed this. Please reformulate contribution 1 (if exists) to focus on the technical contribution based on the targeted scenario. 

5. I have the impression that the definition of localization in this paper is not the common localization problem in drones. But I might be wrong. I also found a description that the leader is positioned based on the neighbors, but then the UAV followers follow the leader for the best position. Which one is the case in this study?

6. In Figure 2, there is an error in P best, and under vi, there is a green line. 

7. Can we make sure the elements in Figure 4 are consistent with the drawing from previous Figures?. 

8. Please reformulate "the system consists of four UAVS.." to "the simulated system consists of.." Btw why later we have 64 UAVs? I am a bit confused by the current description. 

9. What is EOS.IO protocol? this has not been elaborated on in the paper or not referred to anywhere.

10. What does the following phrase mean? "...as UAVs are not affected by the number of leader-UAV"?

11. I don't get Figure 5. It is the 3-D positioning. What are the red buttons? if we have 64UAVs, how do I read the graph as there are obviously more that 64 red dots? How does it read directed follower towards leader? What is the final results here? Put a green button as the leader in the legend.

12. I also have difficulty understanding Figure 6. You need to define the first graph as the initial position, and the second as the position after which time t/which iteration?  the final one is the final position after time t+x or after 50 iterations.

13. How do you compute localization error? 

14. How do you conclude the "increasing number of nodes would minimize the iteration and time consumption to get the target node" and There are also some unpredictable values in (18-20) iteration for several nodes that cause them to miss their target and re-correct their location to return to the time-consuming global swarm path". We don't have any results comparing the different number of nodes vs iteration and time consumption. 

15. I cannot comment much on the security using Blockchain aspects as I don't see any concrete results apart from the pre-determined time to convert a UAV to a leader UAV as 6,47 seconds. The rest I believe, are just proposed schemes. How do we know the technology is safe/robust enough? fast enough? what kind of energy consumption it produces etc. 

Last but not least, please get proofreading, if not professional, at least from the co-authors. A lot of glaring English mistakes, especially in phrasing, but also some typos. 

examples: ..The 5G has recently completed studies.., ..can use to provide..., ..people\.., ..the holes in these studies.., 

Thank you, 

Author Response

(The authors gave the same response as above.)

Round 2

Reviewer 2 Report

First of all, very difficult to see the changes in the manuscript based on your claim, as only several parts got a red text or highlighted in yellow. The authors also didn’t mention where the improvement happens in the letter to the reviewer.

Point 1: First of all I think the title is misleading. The authors' work just considers the 5G signal booster to imagine the use of swarm drones, but the authors didn't include any of the communication aspects (5G parameters and requirements, booster/repeater technical capacity and demand, etc) in the localization and data exchange. As I see it, the proposed schemes can work in any network and cellular technology. So I propose revising the title, amongst others, to include the word "swarm drones" as it is one of the cosu of the study.

Response 1: Thank you very much for your kind feedback. Considering your kind instruction, we have made relevant changes.

Further phrasal improvement proposition: Efficient and Secure WiFi Signal Booster Via Unmanned Aerial Vehicles Repeater based on Swarm Intelligence Localization and Blockchain.

Point 2: My biggest concern in the study is the introduction and related works. In the introduction and related works, the authors just introduce the diverse use of UAVs but not the concrete research gaps.  In current writing, the ideas of using swarm with SIL and Blockchain technology seem like pop-up ideas and are not justified academically.

Response 2: Thanks for the reviewer’s good evaluation and deep consideration. This part was improved in our revised manuscript.

I am afraid I don’t see this improvement in the introduction.

And the Table is too simplistic, and not aligned with the elaboration in Section 2

Point 3: Authors mention several obstacles to developing UAVs using 5GBS, such as increased interference, but this is not discussed in the obtained results or methodology. The lack of durability is not clear to the readers. Author's also mentioned about "early weather prediction is not adopted", which I don't see brings any value to the work unless discussed.

Response 3: Thanks for the reviewer’s kind suggestion. The increasing obstacles to interference were not discussed because of the nature of the environments as this study focuses on remote environments in which the obstacles to interference are few due to the nature of the region. With regard to early weather forecasting, climate sensors can be added and it will be a good future study, but it is not the topic of our work, as the work focuses on improving the communication strategy between UAVs.

Can you explain the interference and weather non-consideration in the text?

You haven't answered the confusion about what durability is.  Are limited space and environment lack of durability?

Point 4: "Using UAVs equipped with a thermal camera, and signal repeater to identify persons in a specific area and boost the wireless signal in this specific area..." is definitely not the authors' contribution, as many have proposed this. Please reformulate contribution 1 (if exists) to focus on the technical contribution based on the targeted scenario.

Response 4: Thanks for the reviewer’s kind suggestion. Clearly, utilizing UAVs with a thermal camera and signal repeater to assist people in a given region by enhancing wireless signal and securing data transmission using Swarm and Blockchain is our contribution..

Ok. Just improve the formulation of Objective 3 to be consistent with Obj 1 and 2. Propose to write “Secure data transmission using blockchain data decentralization”

Point 5: I have the impression that the definition of localization in this paper is not the common localization problem in drones. But I might be wrong. I also found a description that the leader is positioned based on the neighbors, but then the UAV followers follow the leader for the best position. Which one is the case in this study?

Response 5: Thanks for the reviewer’s good evaluation and deep consideration. UAVs localization procedure is as follows; One of the drones finds the problem and shares its location information as a leader drone to other drones which become follower drones.

ok

Point 6: In Figure 2, there is an error in P best, and under vi, there is a green line..

Response 6: Thanks for the reviewer’s kind remarks. This part was addressed in our revised manuscript..

ok

Point 7: Can we make sure the elements in Figure 4 are consistent with the drawing from previous Figures?

Response 7: We appreciate the reviewer’s kind suggestion. Figure 4 is consistent with previous figures in our revised manuscript.

What I wanted to say, the icons in Block and UAV are different in Figure 4 than in Figures 2 and 3.
If can still be considered for improvement; if not, it is passable.

Point 8: Please reformulate "the system consists of four UAVS.." to "the simulated system consists of.." Btw why later we have 64 UAVs? I am a bit confused by the current description..

Response 8: Thanks for the reviewer’s good evaluation and deep consideration. “a suitable number of nodes to show good resolution after several trials”. We have addressed this point in our revised manuscript.

I still don’t get the simulation scenario.. with 4 vs 64. But If the authors are confident with the improved edition, they can leave it.

Point 9: What is EOS.IO protocol? this has not been elaborated on in the paper or not referred to anywhere...

Response 9: Thank you for your question and related remarks. EOS.IO protocol is one of the Blockchain protocols. We have discussed it in our revised manuscript.

I saw you deleted the EOS.IO. Where is the edition?

Point 10: What does the following phrase mean? "...as UAVs are not affected by the number of leader-UAV"?

Response 10: Thank you for highlighting this point. It means there are no restrictions on the number of drones to be leaders, it could be one or more..

This phrase is still not clear. Please improve as such.

“The number of leader UAVs is not analyzed here, as the <WHAT OF>  UAVs are not affected by the number of leader-UAV.

Point 11: I don't get Figure 5. It is the 3-D positioning. What are the red buttons? if we have 64UAVs, how do I read the graph as there are obviously more that 64 red dots? How does it read directed follower towards leader? What is the final results here? Put a green button as the leader in the legend.

Response 11: Thanks for your useful remarks. It is 3D MATLAB simulations with more than 50 iterations to show leader and follower UAVs and figure 6 shows the execution. This part was also addressed in our revised manuscript.

Ok. Just improve the title of Figure 5 to,

“3D MATLAB simulations diagram to show the position of directed follower-UAVs toward the leader-UAV location. The positions illustrated here are based on 50 iterations.”

Point 12: I also have difficulty understanding Figure 6. You need to define the first graph as the initial position, and the second as the position after which time t/which iteration?  the final one is the final position after time t+x or after 50 iterations.

Response 12: We appreciate your concerns. We have tried our best to address this point in our revised manuscript.

ok

Point 13: How do you compute localization error?

Response 13: Thank you for raising this concern. It depends on the general localization factors where each drone rectifies its position and velocity, depending on the swarm position and velocity.

I saw the description in the text. “The location error is Inversely proportional to increasing 321 the number of nodes where the best General swarm position Gj(t) and the best Personal 322 position Pij(t) values are both contributed to correcting the position of the next follower-323 UAV Xij(t+1).

 Is it possible to write it down in a formula?

Point 14: How do you conclude the "increasing number of nodes would minimize the iteration and time consumption to get the target node" and There are also some unpredictable values in (18-20) iteration for several nodes that cause them to miss their target and re-correct their location to return to the time-consuming global swarm path". We don't have any results comparing the different number of nodes vs iteration and time consumption..

Response 14: Thanks for the reviewer’s good evaluation and deep consideration. Unfortunately, there is no satisfactory answer in this regard, as the answer depends on possibilities and trial and error. However in general, the greater number of drones will reduce the error rate in localization and clustering, as the distances are more close and therefore the correction rate will be better, and this does not prevent the presence of unexpected mutations.

Ok. Was this explained in the analysis?

Point 15: I cannot comment much on the security using Blockchain aspects as I don't see any concrete results apart from the pre-determined time to convert a UAV to a leader UAV as 6,47 seconds. The rest I believe, are just proposed schemes. How do we know the technology is safe/robust enough? fast enough? what kind of energy consumption it produces etc.

Last but not least, please get proofreading, if not professional, at least from the co-authors. A lot of glaring English mistakes, especially in phrasing, but also some typos.

examples: ..The 5G has recently completed studies.., ..can use to provide..., ..people\.., ..the holes in these studies..,..

Response 15: We are indebted to you for highlighting these points. Considering your remarks, we have updated our study to address these points as well.

I still see some phrasal issues. But I hope MDPI editors can help with this.